# Evaluation of ambivalent sexism in Colombia and validation of the ASI and AMI brief scales

**Lizeth Cristina Martínez-Baquero**☯, **Pablo Vallejo-Medina**[ID]☯*

School of Psychology, Fundación Universitaria Konrad Lorenz, Bogotá, Colombia

☯ These authors contributed equally to this work.
* pablo.vallejom@konradlorenz.edu.co

## Abstract

Sexism has implications for people's physical and mental health. Thus, understanding sexism and its prevalence is key to understanding the phenomenon. In the current study, 717 Colombian men and women completed the brief scales of Ambivalent Sexism toward women and men and the Gender Identity Scale. The assessment was conducted using a web-based method. Both scales, as expected, were two-dimensional. Reliability ranged from .83 to .88. Moderate and high correlations were observed with the Gender Identity Scale. Men showed higher levels of hostile and benevolent sexism toward women and benevolent sexism toward men. It was also found that the higher the level of education, the lower the rates of sexism toward men and women. The brief scales were valid and reliable for measuring hostile and benevolent sexism in Colombia.

**Data Availability Statement:** The data are held and will be held in a public repository in this link: https://github.com/pableres/papersexism and https://doi.org/10.5281/zenodo.7467165

## Introduction

Every day, there are, on average, 137 femicides worldwide [1]. Latin America is one of the most violent places for women, with 4,640 femicides in 2019 [2]. While 630 femicides were observed in 2020 in Colombia alone [3], 106 more were reported by February 2021 [4]. Femicide is perhaps the gravest form of discrimination suffered by women. The global gender gap calculated in 2019 was 31.4% and showed the differences between men and women in education, health, work, and politics. It will take 99.5 or 135,6 years to equalize the living conditions of men and women. The COVID-19 pandemic also exacerbated the labor gaps between men and women in most Latin American countries [5]. Only 50% of women of working age are employed, as opposed to 75% of men of working age. Women earn 77% of men's wages. Only 50% of those with university degrees gain access to executive positions [6]. Women spend approximately two-thirds of their free time on domestic activities, whereas men spend only one-third of their free time [7]. The global picture shows a higher number of girls out of school; approximately 16 million will never go to school [8]. Women are better qualified than men at the professional level, although they work and earn less than men [6]. Women tend to choose careers related to caregiving as an extension of the social role assigned to this work. In many cases, women abandon their studies because of childbirth, domestic and caregiving responsibilities, and education inaccessibility due to being prohibited by their families [8,9].

**Funding:** The author(s) received no specific funding for this work.

**Competing interests:** he authors have declared that no competing interests exist.

Worldwide, around 2.5 billion women and girls suffer the consequences of discriminatory laws and gaps in legal protection. Most states lack legal mechanisms to prevent, protect, and redress women victims of violence and ensure gender equity [10].

The ascription of traditional gender roles in society has implications for men, especially regarding to health, by validating alcohol consumption and violent and risky sexual behaviors as an expression of virility and strength that should characterize men. Men have a shorter life expectancy: 4.4 years less than women. Their Disability Adjusted Life Years (DALYs) burden is three times more than that of women due to work activities involving their exposure to risk factors [11,12]. Worldwide, more men drink and drink in more significant. By 2016, 2.3 million men died from drinking, and 237 million had alcohol use disorders, five times more likely than women to suffer from them [12] Risky sexual behavior is more prevalent among young and adolescent men because they are pressured to have multiple partners due to result of the dominant man stereotype [13].

Gender ideology comprises the beliefs men and women hold about the roles and behaviors that both sexes should maintain concerning paid work, family responsibilities, and other interactions [14]. Gender ideology affects family processes and is directly related to childcare, the division of domestic activities, conflicts, relationship quality, violence against women, work, and economic gains [15]. Two positions are observed in gender ideology: the traditional and the egalitarian. First, the man is considered an authority figure with economic and social power, whereas the woman is considered secondary and is vested with care and reproduction tasks. Second, the equality of roles that men and women can assume [16]. Sexism is closely related to gender inequality and traditional gender ideologies. It is a set of individuals' attitudes, beliefs, behaviors, and organizational, institutional, and cultural practices that express negative evaluations of individuals based on their gender, causing and maintaining inequality between women and men [17]. In addition, prejudices like sexism and racism share common aspects: discrimination denial, the antagonism between the demands made by groups, and the resentment provoked toward the support policies that some obtain in favor of their rights [18].

Glick and Fiske's [16] theory of ambivalent sexism considers sexism a prejudice marked by deep ambivalence and multidimensionality, denoting a mixture of hostile and benevolent attitudes. Hostile Sexism (HS) is traditional sexism oriented toward the perception of inferiority of the other sex. Benevolent Sexism (BS) presents sexist attitudes with a positive affective tone that allows men to behave pro-socially. However, it helps justify and maintain gender inequality [19]. Both forms of sexism (HS and BS) are based on standard biological and social conditions. Men have economic, legal, and political power, and women are assigned to handle fertility and care. Sexist attitudes can develop in childhood or later developmental stages, and culture favors their acquisition [16,19–21]. The effects of BS and HS have been corroborated in close relationships between men and women. Sexism enables the permeation of sexual violence, beauty ideals and practices, poor self-esteem among women, and limitations on career decisions and aspirations [22].

Gender, age, and religious beliefs can affect sexist attitudes [23–27]. Living in traditional societies with a lower quality of life has also been associated with sexism [28]. Positive associations were observed between sexism and self-concept in men [29], and between life satisfaction and BS, negative associations were observed between life satisfaction and HS [30]. The relationship between romantic attachment and sexism points to attachment avoidance among men as a predisposition toward endorsing HS and rejecting BS [31]. The cooperative and passive styles of interpersonal conflict resolution between men and women correlate positively with BS, while the aggressive style correlates with HS [32]. Sexism is associated with positive attitudes toward rape [33] and psychopathic traits in men [34].

The Ambivalent Sexism Inventory (ASI) and the Ambivalence Toward Men Inventory (AMI), developed by Glik and Fiske in 1996 and 1999, are usually used to measure ambivalent sexism. Two common factors, SH and SB, were identified in both scales. Both the ASI and AMI have short versions, which have demonstrated good psychometric properties and are consistent with the original dimensionality of the scales [35]. Reliability ranges between .72 and .90 for ASI [19,36–46] and between .67 and .85 for AMI [35,47,48]. AMI has received less psychometric attention. ASI and AMI have been used in multiple studies the world over. They have been translated (although not always validated) into French [37], Portuguese [42], Italian [35,41,49], German [38], Polish [40], Hebrew [19,41], Dutch [19], Korean [19], Japanese [19], Norwegian [48], Turkish [45], and Spanish [39], among other languages. In Latin America, they have been widely employed and validated in some countries, given the vital cultural and social component underlying sexism [36,39,43,46,47]. Therefore, the initially proposed two-dimensionality did not always fits. In Colombia the full version of the ASI [19,28,50–53] and the brief version [54] have been previously employed. However, there are no data on how the items have been adapted, their psychometric properties in Colombia, or whether the version from Spain is even comprehensible in Colombia. Although it has been used, it has not yet been validated in Colombia. Therefore, given the cultural differences between Spain and Colombia, the accuracy of the measure in Colombia is questionable.

On the other hand, the AMI, both the brief and the complete version, has not been used in previous studies nor validated in Colombia. Therefore, it is necessary to evaluate the psychometric properties of the short scales of the ASI and the AMI to be able to use them with guarantees in subsequent studies. Furthermore, given the abrupt gender inequalities in Colombia, it would be necessary to have a scale that allows the correct evaluation of sexism. Therefore, the present study aims to validate the short versions of the ASI and AMI scales with the Colombian population and conduct a preliminary analysis of sexism in Colombia.

## Material and methods

### Sample

A total of 717 Colombian adults from approximately 120 Colombian municipalities, aged between 18 and 93 years participated in the study. A total of 29.1% were men, and 70.9% were women. The mean age was 41 years ($SD$ = 13.69). In terms of gender, 27.3% self-identified as cis men, 69.9% as cis women, 0.3% as trans women, 0.3% as trans men, 0.44% as queer, and 1.7% expressed being of another gender. The mean age of women was 40.14 years ($SD$ = 13.00), and that of men was 43.19 ($SD$ = 15.03) years without statistically significant differences t(713) = 2.71; $p$ = .07. The inclusion criteria were being Colombian, aged over 18 years, being able to read and write and sign the informed consent. Subjects of foreign nationality living in Colombia were excluded. Participants were asked to sign an informed consent form and answer all questions. Table 1 presents a summary of the sociodemographic information of the participants. The sample was divided into two halves comprising 358 and 359 adults each. The first group underwent the Exploratory Factor Analysis (EFA), and the second the Confirmatory Factor Analysis (CFA).

### Instruments

ASI-Brief (Brief Ambivalent Sexism Inventory; [55] composed of 12 items. Six items form the Hostile Sexism (HS) subscale, and six items form the Benevolent Sexism (BS) subscale. Scores were obtained using a Likert scale ranging from 0 = strongly disagree to 5 = strongly agree. Higher scores indicate a greater presence of sexism toward women. The brief version was obtained from the complete ASI [16].

**Table 1. Sociodemographic description of participants.**

| Characteristics | Men (n = 207) | Women (n = 506) | Total (n = 717) |
|---|---|---|---|
| Sexual orientation | | | |
| 1. Heterosexual exclusive | 74.90% | 81.40% | 79.50% |
| 2. Mainly heterosexual attraction, with sporadic homosexual contact. | 4.30% | 10.30% | 8.60% |
| 3. Mainly heterosexual attraction, with several homosexual contacts. | 1.90% | 1.60% | 1.70% |
| 4. Bisexual | 1.00% | 1.60% | 1.40% |
| 5. Mainly homosexual attraction, with several heterosexual contacts. | 2.90% | .40% | 1.10% |
| 6. Mainly homosexual attraction, with some sporadic heterosexual contact. | 1.40% | 1.60% | 1.50% |
| 7. Homosexual exclusive | 12.10% | 1.00% | 4,20% |
| 8. Asexual (no interest in sexual contact). | 1.40% | 2.20% | 2.00% |
| 9. Does not report | | | |
| Schooling in years | | | |
| 5 | .5% | .8% | .7% |
| 11 | 7.7% | 8.9% | 8.5% |
| 12 | 8.7% | 8.3% | 8.4% |
| 13 | 10.1% | 4.1% | 5.9% |
| 14 | 9.1% | 9.1% | 9.1% |
| 16 | 27.4% | 26.0% | 26.4% |
| 17 | 9.1% | 7.7% | 8.1% |
| More than 17 | 27.4% | 35.2% | 33.0% |
| Couple | | | |
| Yes | 54.3% | 60.1% | 58.4% |
| No | 45.7% | 39.9% | 41.6% |
| Marital Status | | | |
| Married | 18.8% | 23.2% | 21.9% |
| Single | 52.7% | 45.2% | 47.3% |
| Widowed | .5% | 2.6% | 2.0% |
| Unmarried | 15.0% | 16.5% | 16.1% |
| Separated | 13.0% | 12.6% | 12.7% |

AMI-Brief (Brief Ambivalent Sexism Inventory) [56] is composed of 12 items. Composed of two subscales: 6 items for Hostile Sexism (HS) and 6 for Benevolent Sexism (HS). Scores were obtained using a Likert scale ranging from 0 = strongly disagree to 5 = strongly agree. Higher scores indicate higher levels of sexism toward men. The brief version was based on the original AMI [56].

The Abbreviated Gender Ideology Scale (EIG) [57] measures the traditional conception of gender ideology. It comprises 12 items answered on a Likert-type scale of 9 alternatives (from 1 = totally agree to 10 = totally disagree). In the present study, the Colombian validation [58] comprising 9 items was used. Higher scores show a better conception toward women, indicating a more egalitarian state of roles. The internal consistency for the scale for this study was .82.

Sociodemographic data, participants were asked about their age, gender, sex, and level of schooling.

## Procedure

Before applying the ASI and AMI, a linguistic adjustment process was performed for the two scales following the Guidelines for Translating and Adapting Tests [59] and the

recommendations of AERA, APA, and NCME (2014). We also followed the guidelines for adapting tests from one culture to another within the same language [60]. Thus, five pairs of Colombian psychologists who had lived in Spain for over six months reviewed the instruments and selected some words from the items that, in their estimation, Colombian people may not understand clearly. Once adjustments were addressed, the online survey was created and disseminated (the scales in their Colombian adaptation are found in the appendix).

The evaluation was carried out online using Survey Monkey. The survey was disseminated via Facebook between October 15 and November 27, 2020, through a boosted post to people who met the inclusion criteria. A total amount of USD 200 was spent on the dissemination. All participants gave explicit consent to participate in the study before completing the survey. The Institutional Bioethics Committee of Fundación Universitaria Konrad Lorenz approved the study in compliance with national and international ethical standards for human research. The present study was performed per the ethical standards outlined in the 1964 Declaration of Helsinki and its later amendments. All participants gave explicit written consent to participate in the study. The participants completed the surveys anonymously. A total of 928 questionnaires were returned to database. Of these, 211 were eliminated because 72 did not answer some questions, 17 were non-Colombian, and 122 did not answer the instrument altogether. Thus, the final sample comprised 717 participants.

### Data analysis

The factors extracted in the EFA were obtained using 14 different methods all of them were adequate four the test/sample characteristics. And included: Optimal Coordinates, Parallel Analysis, Kaiser criterion, SE Scree, $R^2$, VSS complexity 2, Velicer's MAP, and BIC, among others. To set the number, both a consensus in most methods and the quality of the indicators were consider. The Optimal Coordinates and Parallel Analysis serve as the current standards. The polychoric matrix was used for the EFA of both ASI and AMI, and Oblimin rotation was applied [61] on a maximum likelihood (ML) estimation method. Factor weights > .30 were considered adequate.

For the CFA, the polychoric matrix was used. The estimation method employed was the Weighted Least Square Mean and Variance Adjusted (WLSMV), given the non-compliance with multivariate normality. The goodness of fit for the measurement models was evaluated using absolute, global, and incremental fit indices, namely the Root Mean Square Error Approximation (RMSEA) and its 90% confidence interval, Comparative Fit Index (CFI), and Tucker-Lewis Index (TLI). Values $\geq$ .95 for CFI and TLI and $\leq$ .06 for RMSEA are considered to represent good model fit [62].

All analyses were performed with R [63] using the RStudio interface [64]. The syntax and data frame can be consulted in this link: YYY (they will be shared in the repository for consultation once the article is accepted). In addition, the use of the following packages is highlighted: parameters, ggplot2, psych, psychometric, lavaan, semPlot, and semTools [65–71].

### Results

We first present the construct validity of the ASI and AMI separately. Next, we present the results concerning psychometric properties, reliability, and criterion validity of both scales. Finally, comparisons of schooling and gender for the SH and SB for man and woman are presented.

## Construct validity of the ASI

Among the methods for estimating the number of factors (AFE), 33.33% defended both the uni-dimensionality and the bi-dimensionality of the scale. The two-dimensional structure was supported by the most robust methods (Optimal Coordinates, Parallel Analysis, Kaiser criterion, SE Scree, R2, and BIC) and the theoretical framework on which the scale was designed. This two-dimensional model explained 59% of the variance. Fig 1 shows the factorial weights obtained for each of the two factors.

After exploring the ASI's two-dimensionality, we confirmed it with the CFA using the second independent subsample (n = 356). The fit of the one- and two-factor models (both independent and related) were compared. Table 2 shows the fit values for each. The two-factor related model had the best fit to the data. Fig 2 shows the standardized solution for the two-factor related model. A covariation of .69 was found between SH and SB. The item weights ranged from .60 to .91 for the SB factor and from .59 to .79 for the SH factor.

## Construct validity of the AMI

We sought to identify the number of factors that comprised the AMI employing the AFE. Six estimation methods proposed two-dimensionality (Bentler, Acceleration factor, SE Scree, R2, VSS complexity 1, and Velicer's MAP); 26.09% of the 23 were tested. The single dimension option was supported by 4 of the 23 methods that were tested (t, p, TLI, RMSEA). The 3-factor model supported five methods: CNG, Optimal Coordinates, Parallel analysis, Kaiser criterion, and VSS complexity 2. Of these, the unifactorial one only showed items 1, 2, 3, 6, and 7 with weights above .30. The 3-factor model isolated items 11 and 12 into a single factor, which is an inconsistency from a theoretical point of view (Fig 3). On the other hand, the two-factor model conformed well to the theory (Fig 4). This two-dimensional structure explained 54% of the variance. We thus proceeded to confirm the dimensionality of the scale.

We carried out CFA with the second subsample (n = 357) in which the fits of three models were compared, namely the one, two independent, and two related factor models. Table 3 shows the fit values for each. The model with two related factors had the best fit indexes. Although the RMSEA was weak, the rest of the indicators were acceptable.

## Some psychometric properties of the items and reliability of the ASI and AMI

The psychometric properties of the items of both the ASI and the AMI were adequate. The items followed a distribution close to normality, although some skewness and kurtosis indices exceeded the value of |1|. The items had good variability with SD between 1 and 2, with corrected item-total correlations consistently higher than .50 and alphas if the item was eliminated, which in no case improved the alpha of its corresponding subscale. Finally, the ordinal alpha was adequate in all subscales. For more information, see Table 4.

**ASI and AMI criterion validity.** Table 5 shows the values obtained for the correlations of SH and SB with the Gender Ideology Scale (GIS). All correlations between the sexism variables and the EIG were significant or either moderate or high, as expected. Furthermore, age also seems to have a low to medium, positive, and significant correlation with all the sexism here assessed. For men: (ASI H = . 22\*\*; ASI B = .40\*\*; AMI H = .25\*\*; AMI B = .24\*\*) and for women: (ASI H = . 17\*\*; ASI B = .17\*\*; AMI H = .28\*\*; AMI B = .12\*\*) Fig 5 shows the results obtained on the differences between HM and BM by gender. Significant differences with moderate-high effect sizes were found for the different comparisons made in terms of gender. Bayesian calculations confirmed these differences. Men exhibited more hostile and BS toward

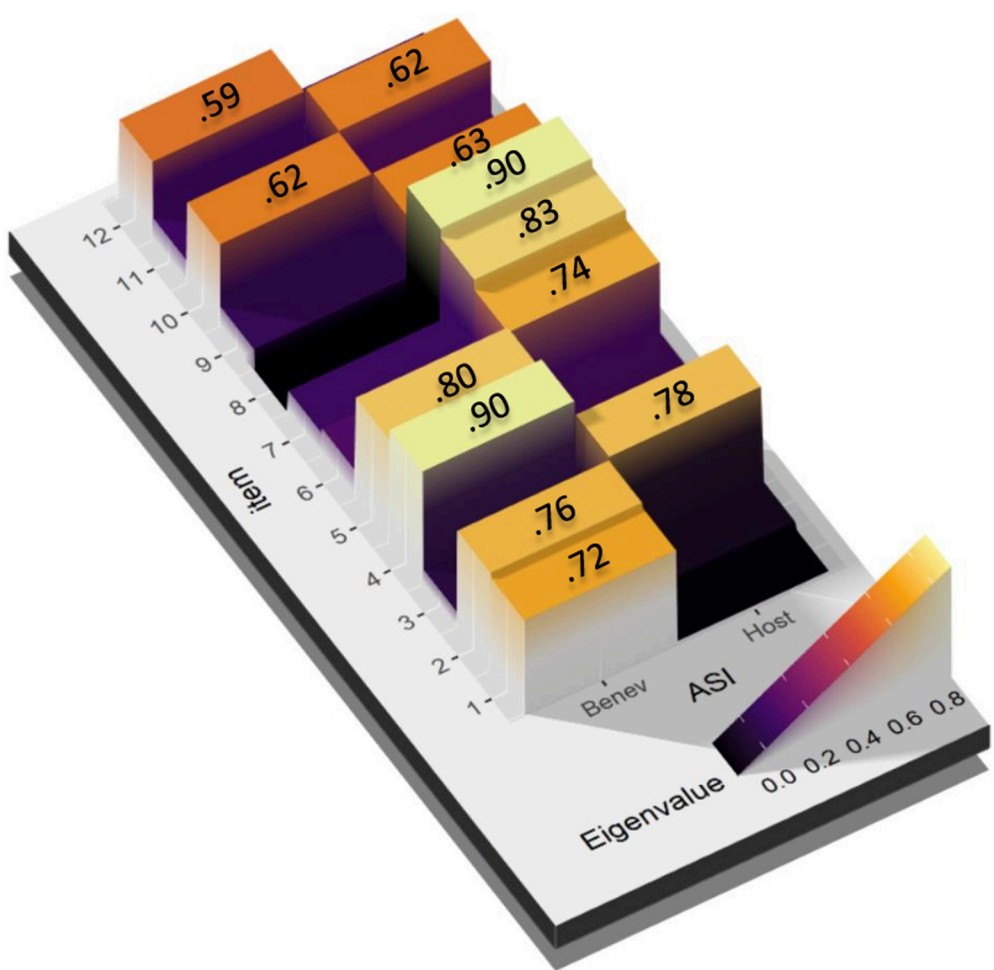

**Fig 1. EFA of the ASI with the eigenvalues for each item.** Factor loadings below .30 have been excluded.

women and scored higher than women in BM. Women were more hostile sexist than men in HM. Statistically significant differences were found concerning to schooling (Fig 6) in all comparisons. However, the observed effect sizes were low. Bayesian calculations also supported this information. The higher the level of schooling, the lower the levels of sexism exhibited by the participants.

## Discussion

This study aimed to validate the brief versions of the ASI and AMI scales with the Colombian population and conduct a preliminary analysis of sexism in Colombia. Both scales showed

**Table 2. Model fit values for ASI.**

| Model | $\chi^2$ | $p$ | gl | TLI | RMSEA | CI 90% | CFI |
|---|---|---|---|---|---|---|---|
| Unifactorial | 617.06 | < .01 | 54 | .77 | .171 | .159 -.183 | .81 |
| Two independent factors | 567.92 | < .01 | 54 | . 79 | .164 | .151-.176 | .83 |
| Two related factors | 167.15 | < .01 | 53 | .95 | .078 | .065-.091 | .96 |

**Note:**df = degree of freedom; TLI = Tucker Lewis Index; RMSEA = Root Mean Square Error Aproximation; CI = Confidence Interval; CFI = Comparative Fit Index.

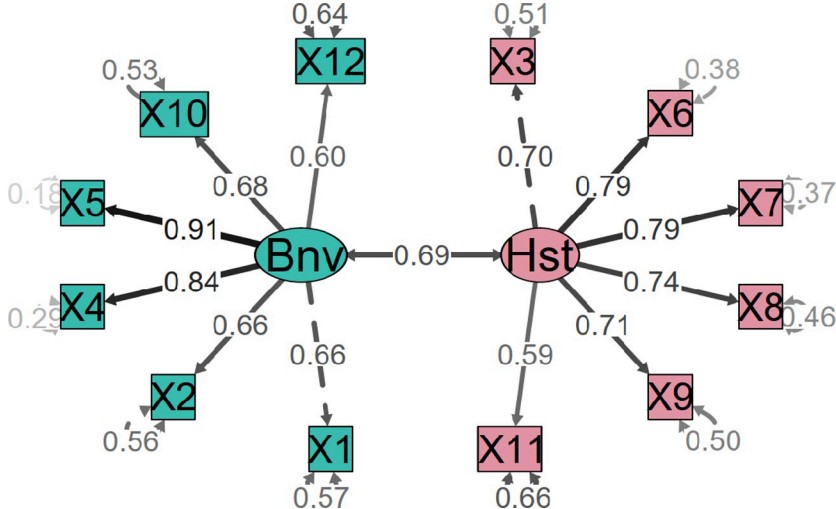

**Fig 2. Path diagram of the CFA ASI.** The standardized weights for the ASI version.

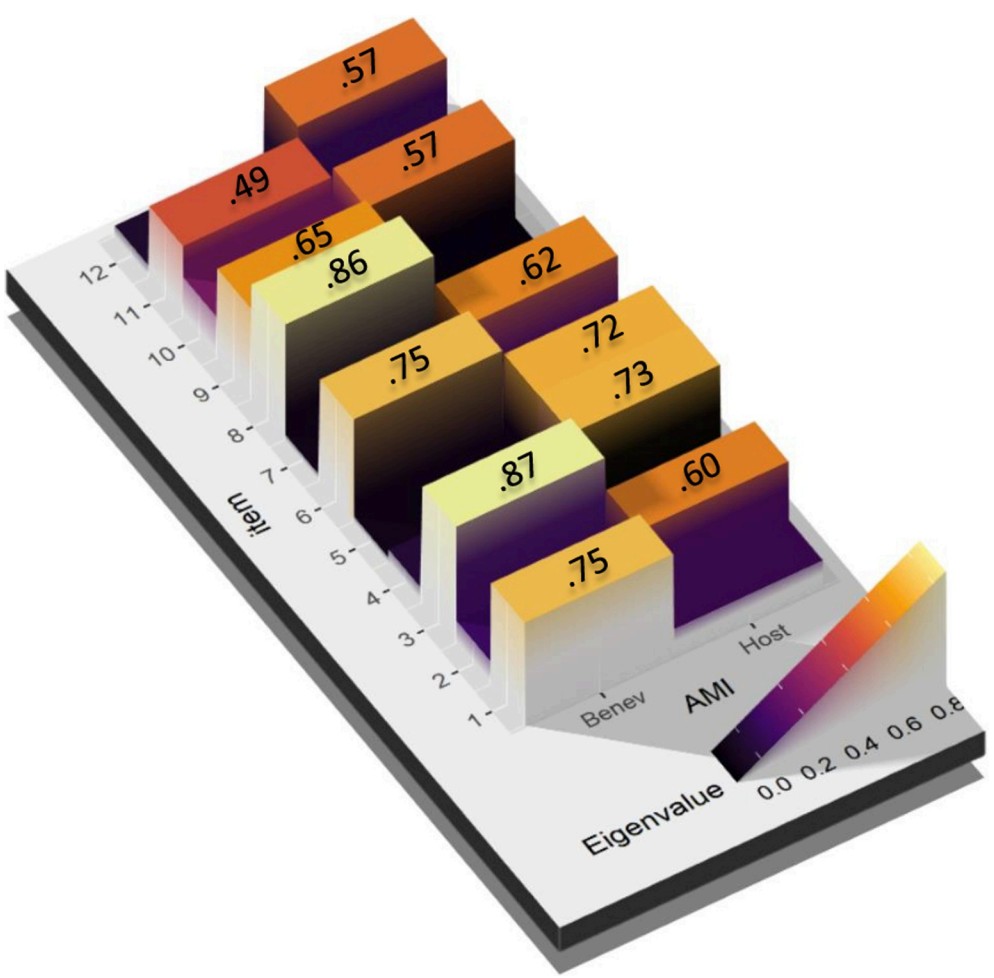

**Fig 3. EFA of the AMI with the eigenvalues for each item.** Factor loadings below .30 have been excluded.

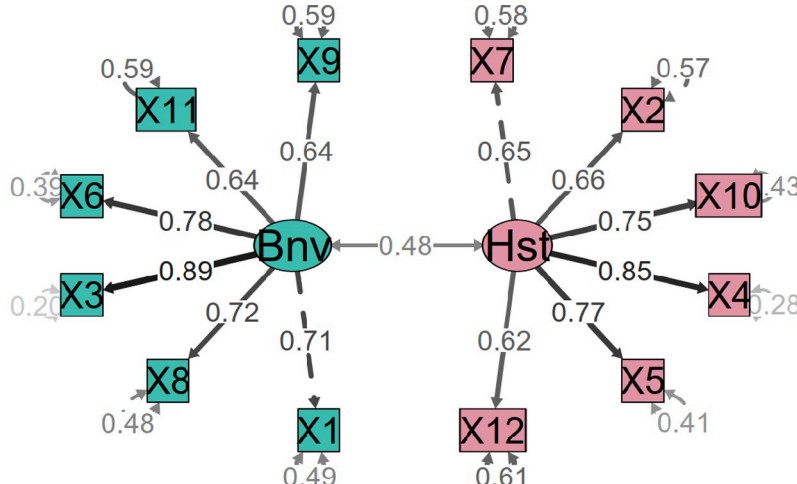

**Fig 4. Path diagram of the CFA AMI.** The standardized weights for the 12-item version of the AMI.

good psychometric properties. The theoretically proposed bifactorial structure [16] was ratified for both scales. It was also found that the higher the level of schooling, the lower the degree of sexism and that man are more sexist than women.

The dimensionality and position of the ASI items in this study coincide with previous studies conducted worldwide [19,35–45,49,55]. However, they do not align with studies conducted using the Spanish versions in Peru [72,73] and Argentina [46] which had identified five factors.

The same dimensionality as expected was also observed about AMI [19,35,40,47] and [41], identified two related second-order factors (SH and SB) that were each divided into three first-order factors. HS comprised paternalism, compensatory gender differentiation, and heterosexual hostility. BS comprised maternalism, complementary gender differentiation, and heterosexual intimacy. The factor analyses in this study showed the distribution of the items in one and two factors (independent and related). This finding coincided with that of [49].

Higher correlations were observed between the Gender Identity Scale with the SB toward both men (AMI) and women (ASI) with respect to traditional gender conceptions. It suggests that the display of traditional gender ideologies was oriented more toward benevolence than hostility [74], representing the presence of people's ascription to traditional gender roles [75]. As expected, this highlighted the existing relationship between traditional and modern forms of sexism [58,75]. A positive correlation between sexism and age was observed in the current study as well as in other studies [26].

Our study showed that men express higher levels of SH toward women, which aligns with the findings of previous studies [40,41,48,49,76]. Men also showed higher SB toward women,

**Table 3. Model fit values for AMI.**

| Model | $\chi^2$ | $p$ | df | TLI | RMSEA | RMSEA CI 90% | CFI |
|---|---|---|---|---|---|---|---|
| Unifactorial | 365.86 | < .01 | 54 | .87 | .128 | .116 - .141 | .90 |
| Two independent factors | 1099.33 | < .01 | 54 | .59 | .235 | .223 - .247 | .66 |
| Two related factors | 117.94 | < .01 | 53 | .97 | .059 | .045 - .073 | .97 |

df = degree of freedom; TLI = Tucker Lewis Index; RMSEA = Root Mean Square Error Aproximation; CI = Confidence Interval; CFI = Comparative Fit Index.

**Table 4. Psychometric properties of the ASI and AMI items and reliability.**

| Scale | Item | Mean | Sd | Skew | Kurtosis | R.drop | Alpha-item | Alpha |
|---|---|---|---|---|---|---|---|---|
| ASI H | 3 | 1.61 | 1.70 | 0.53 | -1.17 | .71 | .86 | .88 |
|  | 6 | 1.25 | 1.54 | 0.94 | -0.39 | .71 | .86 |  |
|  | 7 | 1.74 | 1.68 | 0.43 | -1.20 | .77 | .85 |  |
|  | 8 | 1.59 | 1.61 | 0.60 | -0.89 | .74 | .85 |  |
|  | 9 | 1.77 | 1.67 | 0.40 | -1.20 | .64 | .87 |  |
|  | 11 | 1.98 | 1.80 | 0.31 | -1.32 | .60 | .88 |  |
| ASI B | 1 | 1.80 | 1.72 | 0.44 | -1.18 | .63 | .86 | .87 |
|  | 2 | 2.76 | 1.90 | -0.25 | -1.43 | .65 | .86 |  |
|  | 4 | 1.58 | 1.83 | 0.73 | -0.97 | .78 | .83 |  |
|  | 5 | 1.05 | 1.64 | 1.38 | 0.49 | .78 | .83 |  |
|  | 10 | 1.63 | 1.71 | 0.57 | -1.09 | .58 | .87 |  |
|  | 12 | 0.77 | 1.35 | 1.75 | 2.01 | .64 | .86 |  |
| AMI H | 2 | 1.46 | 1.62 | 0.73 | -0.80 | .57 | .82 | .83 |
|  | 4 | 2.78 | 1.76 | -0.31 | -1.20 | .69 | .79 |  |
|  | 5 | 2.82 | 1.68 | -0.38 | -1.10 | .68 | .79 |  |
|  | 7 | 2.30 | 1.74 | 0.06 | -1.36 | .57 | .81 |  |
|  | 10 | 2.22 | 1.75 | 0.06 | -1.34 | .58 | .81 |  |
|  | 12 | 2.49 | 1.77 | -0.13 | -1.35 | .56 | .82 |  |
| AMI B | 1 | 0.69 | 1.28 | 1.88 | 2.55 | .79 | .85 | .87 |
|  | 3 | 0.41 | 1.06 | 2.96 | 8.39 | .85 | .83 |  |
|  | 6 | 0.72 | 1.27 | 1.81 | 2.39 | .81 | .84 |  |
|  | 8 | 0.64 | 1.24 | 2.05 | 3.40 | .81 | .84 |  |
|  | 9 | 1.84 | 1.66 | 0.36 | -1.17 | .74 | .86 |  |
|  | 11 | 2.10 | 1.70 | 0.15 | -1.31 | .67 | .87 |  |

H = hostile; B = benevolent; ASI = Ambivalent Sexism Inventory; AMI = Ambibalent Sexism Men Inventory; sd = standard deviation; r.drop = item total correlation corrected; alpha–items = ordinal alpha id item is deleted; alpha = ordinal alpha.

coinciding with the findings of studies from other countries [36,42] but differing from that reported by [19], in which women exhibited higher SB. Our results show that men are ambivalent and exhibit high levels of SB and SH than women. This shows the deep prejudice toward

**Table 5. Means, standard deviations, and correlations with confidence intervals.**

| Variable | M | SD | 1 | 2 | 3 | 4 |
|---|---|---|---|---|---|---|
| 1. ASI H | 20.39 | 9.44 |  |  |  |  |
| 2. ASI B | 22.44 | 10.17 | .43** [.37, .48] |  |  |  |
| 3. AMI H | 20.89 | 6.96 | .28** [.22, .34] | .44** [.39, .49] |  |  |
| 4. AMI B | 16.45 | 6.63 | .42** [.37, .48] | .61** [.57, .65] | .50** [.45, .55] |  |
| 5. GIE | 26.09 | 15.97 | .33** [.26, .39] | .55** [.50, .59] | .43** [.37, .48] | .59** [.54, .63] |

M and SD are used to represent mean and standard deviation, respectively. Values in square brackets indicate the 95% confidence interval for each correlation. The confidence interval is a plausible range of population correlations that could have caused the sample correlation.

** indicates $p < .01$.

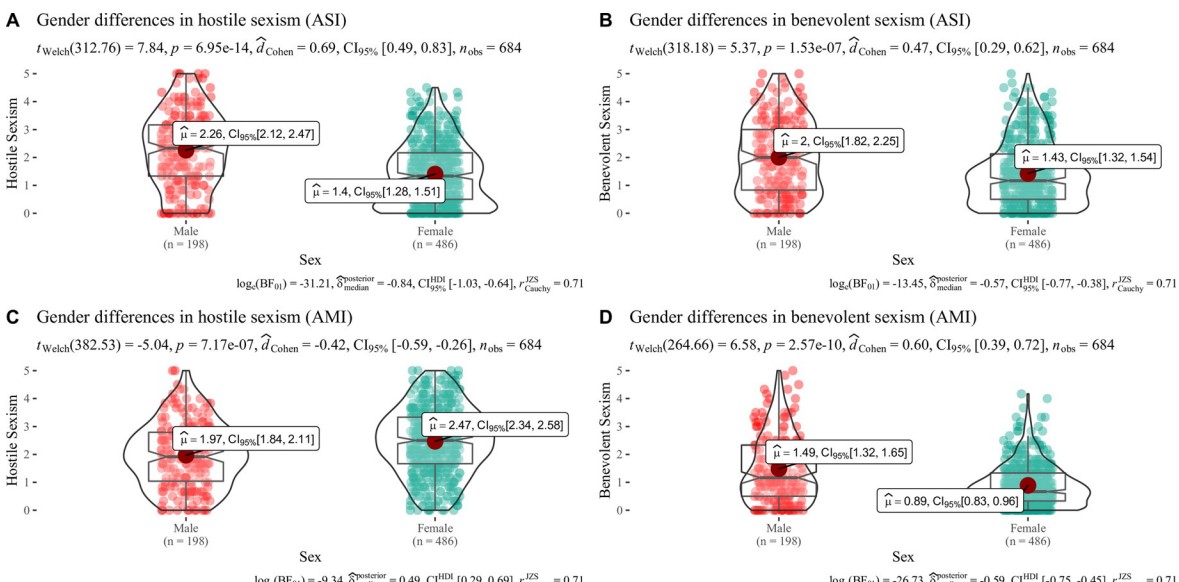

**Fig 5. Differences between sexism.** Box and violin plots representing gender differences between sexism. $\hat{\mu}$ = mean; CI = Confidence Interval; BF = Bayes Factor.

women in Colombia and the lower levels of equality between men and women [41]. If men have a high level of SH toward women, they probably have a strong incentive to accept BS to obtain protection, admiration, and affection from men and thus avoid hostility. Women's attitudes toward men are explained from the perspective of a lower status group, thus affecting the hostility and benevolence they may feel toward men [40]. Our results also show that women are, on average, less benevolent than men toward their in-group and less accepting of

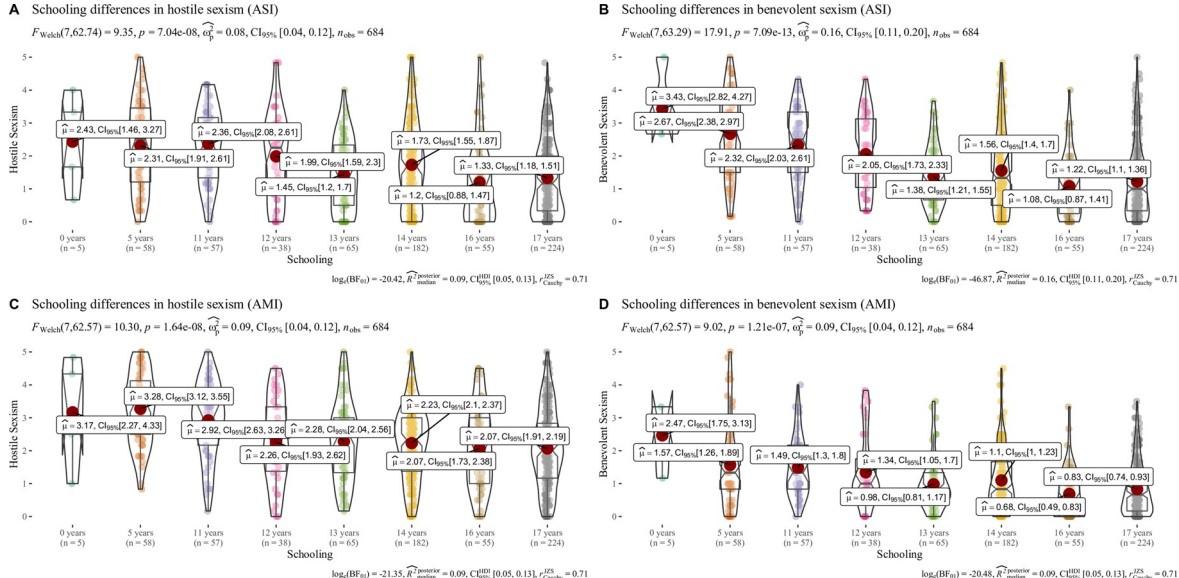

**Fig 6. Schooling differences.** Box and violin plots representing schooling differences between sexism. $\hat{\mu}$ = mean; CI = Confidence Interval; BF = Bayes Factor.

hostility toward their in-group than men, which is consistent with [19]. Women arguably exhibit a less polarized attitude toward their in-group.

We observed a reduction in the rate of sexism in the country compared with [19], in which the scores for SH were 3.1 for men, 2.5 for women, and for SB were 2.7 for men and 2.8 for women. In our study, the scores for SH were 2.6 for men and 1.4 for women, and SB was 2.0 for men and 1.4 for women. Acknowledging that for this study, the sample comprised university students who tend to exhibit lower levels of sexism (both hostile and benevolent), we can gather that the reduction of sexism toward women in Colombia over the last 20 years could be even more significant—placing us in the contemporary context, rather than in the category of the scores of Western nations (United States, United Kingdom, Spain, Australia, and Germany) from two decades ago. However, today's SB scores of Colombian women are lower than those observed in Western countries 20 years ago.

We found that women are more hostile and less benevolent toward men, suggesting a negative evaluation and rejection of traditional gender relations in which women do not accept men's superiority in the non-domestic context [33,40]. This coincides with the findings of previous studies [35,41,56,76]. Men showed a higher level of SB toward their reference group, reflecting their alignment with traditional gender beliefs pertaining to maintaining the role of protectors, providers, and generators of dependency and admiration among women [40]. The only previous study [41]in Colombia with the AMI showed values of HM in men = 2.1, women = 3.0, and BM men = 2.2, women = 2.0. While the present study observed: HM men = 1.9, women = 2.4 and for SB men = 1.4, women = 0.8. Thus, a sexism reduction is observed -especially in the benevolent one- during the last fifteen years. Contrasting the scores obtained here with the US values in HM [41], they are very similar (men = 1.8, women = 2.4), being lower than those exhibited by the Spanish participants (men = 2.4, women = 2.9). Likewise, in terms of BM, the values of this study are lower than those of the American (men = 1.9, women = 1.5) and Spanish (men = 2.1, women = 1.9) participants. Thus, men may be evaluated less positively than women (by both men and women). These differences do not reveal greater gender equality as hostility toward men would be maintained as an inevitable and natural response to their power status. However, it seems that benevolence toward the protective figure of the "weaker sex" is decreasing considerably [41].

Our results indicate that sexism, regardless of whether it is oriented toward women or men, decreases as the number of years of schooling increases. This is consistent with other studies [26,42,77,78] that have identified the role of education in displaying sexist prejudices and attitudes and ascribing to traditional gender roles.

## Conclusion

This study reaffirms the validity and reliability of the brief versions of the ASI and AMI scales as proposed by [56]to measure sexism as a form of prejudice in the Colombian context. The factorial structure and confirmatory analyses supported the original models [16,41,56]. The differences between sexes in the display of SH and SB toward women and men reaffirm the theoretical approaches in which ambivalence in the relationships between the in-groups is established and where sexist attitudes continue to prevail. The role of schooling in the exhibition of sexist prejudices is evident in Colombia.

This study has a few limitations. First, the participants had access to social networks. People without access to this type of service may have been excluded. Although people from different parts of the country participated, some departments that had no infrastructure and were located farther away from the central zone did not. Social desirability as a variable was not controlled for. The use of short scales and the comparison with the averages of the extended scales

should be interpreted with caution. In the future, sexism and its relationship with other variables, like gender violence in all its manifestations, should be studied in greater depth, as research on the subject has been abandoned over the last two decades despite the persistent manifestations of inequality between the sexes.

## Supporting information

**S1 File. Appendix.** Items Spanish and English. In this file are the items in English and Spanish of the short version of ASI and AMI.
(DOCX)

## Author Contributions

**Conceptualization:** Lizeth Cristina Martínez-Baquero, Pablo Vallejo-Medina.

**Data curation:** Lizeth Cristina Martínez-Baquero, Pablo Vallejo-Medina.

**Formal analysis:** Lizeth Cristina Martínez-Baquero, Pablo Vallejo-Medina.

**Investigation:** Lizeth Cristina Martínez-Baquero, Pablo Vallejo-Medina.

**Methodology:** Lizeth Cristina Martínez-Baquero, Pablo Vallejo-Medina.

**Project administration:** Lizeth Cristina Martínez-Baquero, Pablo Vallejo-Medina.

**Resources:** Lizeth Cristina Martínez-Baquero, Pablo Vallejo-Medina.

**Software:** Lizeth Cristina Martínez-Baquero, Pablo Vallejo-Medina.

**Supervision:** Pablo Vallejo-Medina.

**Validation:** Lizeth Cristina Martínez-Baquero, Pablo Vallejo-Medina.

**Visualization:** Lizeth Cristina Martínez-Baquero, Pablo Vallejo-Medina.

**Writing – original draft:** Lizeth Cristina Martínez-Baquero, Pablo Vallejo-Medina.

**Writing – review & editing:** Lizeth Cristina Martínez-Baquero, Pablo Vallejo-Medina.

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
