## [Decision Letter · Decision Letter 0]

5 Oct 2022

PONE-D-22-15490

Evaluation of Ambivalent Sexism in Colombia and Validation of the ASI and AMI Brief Scales

PLOS ONE

Dear Dr Hugo Sanchez Hernandez, M.A.

Thank you for submitting your manuscript to PLOS ONE. After careful consideration, we feel that it has merit but does not fully meet PLOS ONE’s publication criteria as it currently stands. Therefore, we invite you to submit a revised version of the manuscript that addresses the points raised during the review process.

**Comments to the Author**

1. Is the manuscript technically sound, and do the data support the conclusions?

Reviewer #1: Yes

Reviewer #2: Yes

2. Has the statistical analysis been performed appropriately and rigorously?

Reviewer #1: Yes

Reviewer #2: Yes

3. Have the authors made all data underlying the findings in their manuscript fully available?

Reviewer #1: Yes

Reviewer #2: Yes

4. Is the manuscript presented in an intelligible fashion and written in standard English?

Reviewer #1: Yes

Reviewer #2: Yes

5. Review Comments to the Author

Reviewer #1: I think it is a good paper with the following few exceptions:

1. There are minor language editorial issues that should be looked into., maybe get a proofreader.

2. Page 3 has an author's name and year, is that the Vancouver excerption when it comes to discussing a theoretical framework?

3. I suggest that the supporting information/appendices that appear at the end of the paper be moved either to the body of the text where there are discussions about the figures or before the bibliography so that the bibliography is the last section.

4. I also suggest that for multiple sources which appear in-text, a comma is used to separate them instead of consecutive multiple brackets as seen for example at the top of page 18.

5. If possible, include a conclusion heading.

Reviewer #2: The title has been outlined together with the forms (HS and BS). Consider checking under the instruments heading the ASI-brief and the AMI-brief share the same paragraphs.

Too much spacing in between the heading and subheading, consider removing the extra spaces and lines.

We look forward to receiving your revised manuscript.

Kind regards,

Malesela Edward Eddie Montle, PhD

Academic Editor

PLOS ONE

Journal Requirements:

2. Please change "female” or "male" to "woman” or "man" as appropriate, when used as a noun (see for instance https://apastyle.apa.org/style-grammar-guidelines/bias-free-language/gender).

3. Please ensure that you include a title page within your main document. We do appreciate that you have a title page document uploaded as a separate file, however, as per our author guidelines (http://journals.plos.org/plosone/s/submission-guidelines#loc-title-page) we do require this to be part of the manuscript file itself and not uploaded separately.

Unfunded studies

Enter: The author(s) received no specific funding for this work.

NO authors have competing interests

Enter: The authors have declared that no competing interests exist.

6. Please ensure that you include a title page within your main document. You should list all authors and all affiliations as per our author instructions and clearly indicate the corresponding author.

**Comments to the Author**

1. Is the manuscript technically sound, and do the data support the conclusions?

Reviewer #1: Yes

Reviewer #2: Yes

2. Has the statistical analysis been performed appropriately and rigorously?

Reviewer #1: Yes

Reviewer #2: Yes

3. Have the authors made all data underlying the findings in their manuscript fully available?

Reviewer #1: Yes

Reviewer #2: Yes

4. Is the manuscript presented in an intelligible fashion and written in standard English?

Reviewer #1: Yes

Reviewer #2: Yes

5. Review Comments to the Author

Reviewer #1: I think it is a good paper with the following few excerptions:

1. There are minor language editorial issues that should be looked into., maybe get a proof reader.

2. Page 3 has an author's name and year, is that the Vancouver excerption when it comes to discussing a theoretical framework?

3. I suggest that the supporting information/appendices that appear at the end of the paper be moved either to the body of the text where there are discussions about the figures or before the bibliography so that the bibliography is the last section.

4. I also suggest that for multiple sources which appear in-text, a comma is used to separate them instead of consecutive multiple brackets as seen for example at the top of page 18.

5. If possible, include a conclusion heading.

Reviewer #2: The title has been outlined together with the forms (HS and BS). Consider checking under the instruments heading the ASI-brief and the AMI-brief share the same paragraphs.

Too much spacing in between the heading and subheading, consider removing the extra spaces and lines.

PONE-D-22-15490

Evaluation of Ambivalent Sexism in Colombia and Validation of the ASI and AMI Brief Scales

PLOS ONE

Dear Dr Hugo Sanchez Hernandez, M.A.

Thank you for submitting your manuscript to PLOS ONE. After careful consideration, we feel that it has merit but does not fully meet PLOS ONE’s publication criteria as it currently stands. Therefore, we invite you to submit a revised version of the manuscript that addresses the points raised during the review process.

**Comments to the Author**

1. Is the manuscript technically sound, and do the data support the conclusions?

Reviewer #1: Yes

Reviewer #2: Yes

2. Has the statistical analysis been performed appropriately and rigorously?

Reviewer #1: Yes

Reviewer #2: Yes

3. Have the authors made all data underlying the findings in their manuscript fully available?

Reviewer #1: Yes

Reviewer #2: Yes

4. Is the manuscript presented in an intelligible fashion and written in standard English?

Reviewer #1: Yes

Reviewer #2: Yes

5. Review Comments to the Author

Reviewer #1: I think it is a good paper with the following few excerptions:

1. There are minor language editorial issues that should be looked into., maybe get a proof reader.

2. Page 3 has an author's name and year, is that the Vancouver excerption when it comes to discussing a theoretical framework?

3. I suggest that the supporting information/appendices that appear at the end of the paper be moved either to the body of the text where there are discussions about the figures or before the bibliography so that the bibliography is the last section.

4. I also suggest that for multiple sources which appear in-text, a comma is used to separate them instead of consecutive multiple brackets as seen for example at the top of page 18.

5. If possible, include a conclusion heading.

Reviewer #2: The title has been outlined together with the forms (HS and BS). Consider checking under the instruments heading the ASI-brief and the AMI-brief share the same paragraphs.

Too much spacing in between the heading and subheading, consider removing the extra spaces and lines.

We look forward to receiving your revised manuscript.

Kind regards,

Malesela Edward Eddie Montle, PhD

Academic Editor

PLOS ONE

Journal Requirements:

**Comments to the Author**

1. Is the manuscript technically sound, and do the data support the conclusions?

Reviewer #1: Yes

Reviewer #2: Yes

2. Has the statistical analysis been performed appropriately and rigorously?

Reviewer #1: Yes

Reviewer #2: Yes

3. Have the authors made all data underlying the findings in their manuscript fully available?

Reviewer #1: Yes

Reviewer #2: Yes

4. Is the manuscript presented in an intelligible fashion and written in standard English?

Reviewer #1: Yes

Reviewer #2: Yes

5. Review Comments to the Author

Reviewer #1: I think it is a good paper with the following few excerptions:

1. There are minor language editorial issues that should be looked into., maybe get a proof reader.

2. Page 3 has an author's name and year, is that the Vancouver excerption when it comes to discussing a theoretical framework?

3. I suggest that the supporting information/appendices that appear at the end of the paper be moved either to the body of the text where there are discussions about the figures or before the bibliography so that the bibliography is the last section.

4. I also suggest that for multiple sources which appear in-text, a comma is used to separate them instead of consecutive multiple brackets as seen for example at the top of page 18.

5. If possible, include a conclusion heading.

Reviewer #2: The title has been outlined together with the forms (HS and BS). Consider checking under the instruments heading the ASI-brief and the AMI-brief share the same paragraphs.

Too much spacing in between the heading and subheading, consider removing the extra spaces and lines.

---

## [Author Response · Author response to Decision Letter 0]

1 Jan 2023

Bogotá, November 4th 2022

Response to Reviewers

Thanks for the suggestions and comments to improve the quality of our article. All comments were considered and the adjustment was realized.

Reviewer #1:

1. There are minor language editorial issues that should be looked into., maybe get a proofreader.

Authors comments: The document was completed reviewed and adjusted.

2. Page 3 has an author's name and year, is that the Vancouver excerption when it comes to discussing a theoretical framework?

Authors' comments: The mention of the Ambivalent Sexism Theory authors is important for the main idea of the paragraph. We adjusted the citation to the Vancouver format.

3. I suggest that the supporting information/appendices that appear at the end of the paper be moved either to the body of the text where there are discussions about the figures or before the bibliography so that the bibliography is the last section.

Authors' comments: We accept the suggestion. We included the Appendix before the bibliography.

4. I also suggest that for multiple sources which appear in-text, a comma is used to separate them instead of consecutive multiple brackets as seen for example at the top of page 18.

Authors' comments: We adjusted the multiple cites like the reviewer's recommendation.

5. If possible, include a conclusion heading.

Authors comments: We included the subtitle: “conclusion”.

Reviewer #2: 

1. The title has been outlined together with the forms (HS and BS). Consider checking under the instruments heading that the ASI-brief and the AMI-brief share the same paragraphs.

Authors' comments: We reviewed and made them and made.

2. Too much spacing between the heading and subheading, consider removing the extra spaces and lines.

Authors' comments: We checked them and did them 

Additional Adjusts:

1. 

2. We checked them and did them.

3. We included a title page within our main document.

4.4 Unfunded studies: “The authors received no specific funding for this work.”

5.5 Competing Interests: “The authors have declared that no competing interests exist." 

6.6 We included a title page within our main document.

7.7 Done

8.8 Done

I'd like to thank you for your valuable review.

Yours truly,

Lizeth Cristina Martínez-Baquero

PhD. Student

PhD. Pablo Vallejo Medina

---

## [Decision Letter · Decision Letter 1]

19 Jul 2023

PONE-D-22-15490R1Evaluation of Ambivalent Sexism in Colombia and Validation of the ASI and AMI Brief ScalesPLOS ONE

Dear Dr.Vallejo-Medina ,

Thank you for submitting your manuscript to PLOS ONE. After careful consideration, we feel that it has merit but does not fully meet PLOS ONE’s publication criteria as it currently stands. Therefore, we invite you to submit a revised version of the manuscript that addresses the points raised during the review process.

Please make the revisions to the methods and discussion section as requested by Reviewer 3

We look forward to receiving your revised manuscript.

Kind regards,

Rosemary Frey

Academic Editor

PLOS ONE

Journal Requirements:

Reviewers' comments:

Reviewer's Responses to Questions

**Comments to the Author**

1. If the authors have adequately addressed your comments raised in a previous round of review and you feel that this manuscript is now acceptable for publication, you may indicate that here to bypass the “Comments to the Author” section, enter your conflict of interest statement in the “Confidential to Editor” section, and submit your "Accept" recommendation.

Reviewer #2: All comments have been addressed

Reviewer #3: All comments have been addressed

2. Is the manuscript technically sound, and do the data support the conclusions?

Reviewer #2: Yes

Reviewer #3: Yes

3. Has the statistical analysis been performed appropriately and rigorously? 

Reviewer #2: Yes

Reviewer #3: Yes

4. Have the authors made all data underlying the findings in their manuscript fully available?

Reviewer #2: Yes

Reviewer #3: Yes

5. Is the manuscript presented in an intelligible fashion and written in standard English?

Reviewer #2: Yes

Reviewer #3: Yes

6. Review Comments to the Author

Reviewer #2: (No Response)

Reviewer #3: Reviewer Recommendation and Comments for Manuscript Number PONE-D-22-15490R1

Materials and methods

— The inclusion criteria of the participants are very generic

— Explain why there is a division of samples where a first group is subjected to an EFA and the second to a CFA.

— In the factors extracted in the EFA, they were obtained using 24 different methods, clarifying the relevance of these methods given the non-use of others valid for the EFA.

Process

— Specify the linguistic adjustment process for the two scales carried out.

Discussion

— A brief analysis of the results obtained by age, both in men and women, would have been very interesting.

General features

— There are very generic and absolute statements in the text, modify the wording of certain paragraphs such as:

“We found that women are more hostile and less benevolent towards men, indicating a negative evaluation and rejection of traditional gender relations in which women do not accept men's superiority in the non-domestic context”

7. PLOS authors have the option to publish the peer review history of their article (what does this mean?). If published, this will include your full peer review and any attached files.

Reviewer #2: No

Reviewer #3: **Yes: **JOSE LUIS GIL BERMEJO https://orcid.org/0000-0001-6946-5664

---

## [Author Response · Author response to Decision Letter 1]

21 Jul 2023

Reviewer Recommendation and Comments for Manuscript Number PONE-D-22-15490R1

Thank you so much for the commentaries, they really improved the manuscript. 

Materials and methods

— The inclusion criteria of the participants are very generic.

We have included more info.

— Explain why there is a division of samples where a first group is subjected to an EFA and the second to a CFA.

It is not recommended to confirm in a sample what has already been explored in that same sample. Logically, if we test the confirmation of the same model explored in the same sample, the fit will always be almost perfect. We have not added more information in this regard in the article.

-In the factors extracted in the EFA, they were obtained using 24 different methods, clarifying the relevance of these methods given the non-use of others valid for the EFA.

Sorry we test 14 no 24 methods. All of theme were adequate four the test/sample characteristics. Those were: 

Acceleration factor 

CNG 

beta Multiple_regression

VSS complexity 1 

VSS complexity 2 

Velicer's MAP 

Optimal coordinates 

Parallel analysis 

Kaiser criterion 

t Multiple_regression

p Multiple_regression

Scree (R2) 

Scree (SE) 

BIC 

BIC (adjusted) 

Bentler 

Bartlett 

Anderson 

Lawley

We have corrected the erratum and nuanced why we used them.

Process

— Specify the linguistic adjustment process for the two scales carried out.

In the cited paper (60) “Desarrollo de guías para adaptar cuestionarios dentro de una misma lengua en otra cultura” the details can be consulted in both English and Spanish. 

Discussion

— A brief analysis of the results obtained by age, both in men and women, would have been very interesting.

This analysis is now included in the results section and briefly discussed in the discussion section.

General features

— There are very generic and absolute statements in the text, modify the wording of certain paragraphs such as:

“We found that women are more hostile and less benevolent towards men, indicating a negative evaluation and rejection of traditional gender relations in which women do not accept men's superiority in the non-domestic context”

We have modified the wording.

---

## [Decision Letter · Decision Letter 2]

16 Jan 2024

Evaluation of Ambivalent Sexism in Colombia and Validation of the ASI and AMI Brief Scales

PONE-D-22-15490R2

Dear Dr. Vallejo-Medina,

We’re pleased to inform you that your manuscript has been judged scientifically suitable for publication and will be formally accepted for publication once it meets all outstanding technical requirements.

Kind regards,

Rosemary Frey

Academic Editor

PLOS ONE

Additional Editor Comments (optional):

Reviewers' comments:

Reviewer's Responses to Questions

**Comments to the Author**

1. If the authors have adequately addressed your comments raised in a previous round of review and you feel that this manuscript is now acceptable for publication, you may indicate that here to bypass the “Comments to the Author” section, enter your conflict of interest statement in the “Confidential to Editor” section, and submit your "Accept" recommendation.

Reviewer #3: All comments have been addressed

2. Is the manuscript technically sound, and do the data support the conclusions?

Reviewer #3: Yes

3. Has the statistical analysis been performed appropriately and rigorously? 

Reviewer #3: Yes

4. Have the authors made all data underlying the findings in their manuscript fully available?

Reviewer #3: Yes

5. Is the manuscript presented in an intelligible fashion and written in standard English?

Reviewer #3: Yes

6. Review Comments to the Author

Reviewer #3: The improvements made by the authors have substantially enriched the text, giving it a scientific and accessible character for those who access it.

7. PLOS authors have the option to publish the peer review history of their article (what does this mean?). If published, this will include your full peer review and any attached files.

Reviewer #3: **Yes: **JOSE LUIS GIL BERMEJO

---

## [Editor Report · Acceptance letter]

20 Feb 2024

PONE-D-22-15490R2 

PLOS ONE

Dear Dr. Vallejo-Medina, 

I'm pleased to inform you that your manuscript has been deemed suitable for publication in PLOS ONE. Congratulations! Your manuscript is now being handed over to our production team.

Kind regards, 

on behalf of

Dr. Rosemary Frey 

Academic Editor

PLOS ONE